# Neighborhood Deprivation and Racial Disparities in Early Pregnancy Impaired Glucose Tolerance

**DOI:** 10.3390/ijerph20126175

**Published:** 2023-06-19

**Authors:** Cara D. Dolin, Anne M. Mullin, Rachel F. Ledyard, Whitney R. Bender, Eugenia C. South, Celeste P. Durnwald, Heather H. Burris

**Affiliations:** 1Department of Obstetrics and Gynecology, Women’s Health Institute, Cleveland Clinic Lerner College of Medicine, Cleveland, OH 44195, USA; 2Tufts University School of Medicine, Boston, MA 02111, USA; 3Division of Neonatology, Children’s Hospital of Philadelphia, University of Pennsylvania, Philadelphia, PA 19104, USA; 4Department of Obstetrics and Gynecology, Virginia Commonwealth University School of Medicine, Richmond, VA 23298, USA; 5Urban Health Lab, Department of Emergency Medicine, Perelman School of Medicine, University of Pennsylvania, Philadelphia, PA 19104, USA; 6Department of Obstetrics and Gynecology, University of Pennsylvania Perelman School of Medicine, Philadelphia, PA 19104, USA; 7Department of Pediatrics, Perelman School of Medicine, University of Pennsylvania, Philadelphia, PA 19104, USA; 8Leonard Davis Institute, Perelman School of Medicine, University of Pennsylvania, Philadelphia, PA 19104, USA

**Keywords:** census tract, health disparities, maternal obesity, social determinants of health, preconception, prediabetes

## Abstract

Objective: There is mounting evidence that neighborhoods contribute to perinatal health inequity. We aimed (1) to determine whether neighborhood deprivation (a composite marker of area-level poverty, education, and housing) is associated with early pregnancy impaired glucose intolerance (IGT) and pre-pregnancy obesity and (2) to quantify the extent to which neighborhood deprivation may explain racial disparities in IGT and obesity. Study Design: This was a retrospective cohort study of non-diabetic patients with singleton births ≥ 20 weeks’ gestation from 1 January 2017–31 December 2019 in two Philadelphia hospitals. The primary outcome was IGT (HbA1c 5.7–6.4%) at <20 weeks’ gestation. Addresses were geocoded and census tract neighborhood deprivation index (range 0–1, higher indicating more deprivation) was calculated. Mixed-effects logistic regression and causal mediation models adjusted for covariates were used. Results: Of the 10,642 patients who met the inclusion criteria, 49% self-identified as Black, 49% were Medicaid insured, 32% were obese, and 11% had IGT. There were large racial disparities in IGT (16% vs. 3%) and obesity (45% vs. 16%) among Black vs. White patients, respectively (*p* < 0.0001). Mean (SD) neighborhood deprivation was higher among Black (0.55 (0.10)) compared with White patients (0.36 (0.11)) (*p* < 0.0001). Neighborhood deprivation was associated with IGT and obesity in models adjusted for age, insurance, parity, and race (aOR 1.15, 95%CI: 1.07, 1.24 and aOR 1.39, 95%CI: 1.28, 1.52, respectively). Mediation analysis revealed that 6.7% (95%CI: 1.6%, 11.7%) of the Black-White disparity in IGT might be explained by neighborhood deprivation and 13.3% (95%CI: 10.7%, 16.7%) by obesity. Mediation analysis also suggested that 17.4% (95%CI: 12.0%, 22.4%) of the Black–White disparity in obesity may be explained by neighborhood deprivation. Conclusion: Neighborhood deprivation may contribute to early pregnancy IGT and obesity–surrogate markers of periconceptional metabolic health in which there are large racial disparities. Investing in neighborhoods where Black patients live may improve perinatal health equity.

## 1. Introduction

Glycated hemoglobin (HbA1c), the percentage of hemoglobin with attached glucose, represents the average blood glucose during the lifespan of the red blood cells and is used to diagnose diabetes mellitus (HbA1c ≥ 6.5%) and assess glycemic control and response to treatment in non-pregnant adults. A HbA1c of 5.7–6.4% represents impaired glucose tolerance (IGT), or prediabetes, in non-pregnant adults [1]. Racial disparities in HbA1c have been consistently reported, with non-Hispanic Black people having higher HbA1c levels than non-Hispanic White people [2]. In early pregnancy, HbA1c is increasingly being utilized to identify patients with pregestational diabetes. Studies have demonstrated that IGT in early pregnancy is associated with the development of gestational diabetes as well as other adverse pregnancy outcomes including preeclampsia, preterm birth, large-for-gestational age and macrosomia, shoulder dystocia, and cesarean delivery [3,4,5,6]. While early identification of IGT has not been shown to improve maternal and neonatal outcomes, this may be due to early fetal programming and interventions occurring too late in gestation to impact outcomes demonstrating the need to focus on preconception health [7,8].

Early pregnancy and preconception health is likely influenced by many factors, including social and environmental exposures [9]. There is mounting evidence that neighborhood-level exposures such as housing quality, violence, access to grocery stores, and poverty contribute to perinatal health inequity independently of individual-level factors such as education, income, exposure to discrimination, and food insecurity [9,10,11,12,13,14]. This is important because it emphasizes the need to address health inequity at the neighborhood level; interventions that focus solely on individual factors will fall short. Cities across the United States are highly segregated as a result of historical racist redlining, a form of place-based discrimination that deprived neighborhoods where most residents were Black of governmental resources, as well as more recent ongoing real estate practices that have maintained racial segregation and neighborhood inequality [15,16]. Because resources have been distributed inequitably along racial lines, Black people are more likely to be exposed to disadvantaged neighborhood environments which may explain a significant proportion of racial disparities in health that are unaccounted for by individual-level factors [17]. In nonpregnant adults, neighborhood-level poverty and access to nutritious foods have been shown to be associated with adverse health outcomes including diabetes and IGT [18]. There is little known about the role of neighborhood exposures in IGT in early pregnancy. We hypothesized that higher neighborhood deprivation indices would be associated with IGT, and because of ongoing racial residential segregation, differences in neighborhood deprivation would partially explain racial disparities in IGT. 

## 2. Materials and Methods

The study population was derived from the GeoBirth cohort [19,20,21], a retrospective pregnancy cohort of all births in two Penn Medicine hospitals in Philadelphia, PA, United States, that combined account for approximately 9000 births annually. Inclusion criteria for these analyses were the following: a valid residential address within Philadelphia, an HbA1c lab result of <6.5% during the first 20 weeks of pregnancy, a singleton birth ≥ 20 weeks’ gestation between 1 January 2017, and 31 December 2019, and non-missing body mass index (BMI) data (Figure 1). HbA1c lab results were obtained from the electronic health record (EHR) and categorized as normal (<5.7%) or IGT (5.7–6.4%). Demographic and clinical data were also abstracted from the EHR. Race and ethnicity (Hispanic or non-Hispanic) were patient-identified at the time of registration for outpatient visits and inpatient hospitalizations. We categorized race and ethnicity as Non-Hispanic Black (Black), and Non-Hispanic White (White), and due to small numbers, we combined Hispanic, other, multiple, and unknown racial and ethnic designations into a single group. BMI was calculated using height and pre-pregnancy weight from the EHR, if available. If only height was missing, the average height of a female adult was used (64 inches). If pre-pregnancy weight was missing (*n* = 2118), weight at admission for delivery was used and pre-pregnancy weight was calculated using gestational age at delivery and recommended weight gain by week of pregnancy [22]. Obesity was defined as a pre-pregnancy BMI ≥ 30 kg/m^2^. This study was approved by the University of Pennsylvania Institutional Review Board with a waiver of informed consent. 

Neighborhood factors that influence health outcomes can be assessed through composite measures such as deprivation and vulnerability indices [23,24,25,26]. We used a census tract area-level neighborhood deprivation index [23] to quantify the exposure of interest; the index uses American Community Survey 2015 indicators for the median household income, the fraction of households with income below the poverty level, without a high school degree, without health insurance coverage, receiving public assistance income or supplemental nutrition assistance program, and the fraction of houses that are vacant [27]. The neighborhood deprivation index ranges from 0 to 1 with higher values indicating more deprivation. Each patient’s residential address at the time of the HbA1c lab result was geocoded using ArcMAP 10.8, Environmental Systems Research Institute (ESRI), Redlands, CA, and the ArcGIS Street Map Premium North America version 2021.1 address locator with a minimum match score of 75; there were 218 patients excluded because their address was not matched (Figure 1). The resulting coordinates were mapped to the US Census Bureau’s 2019 cartographic boundary shapefile to assign each address a census tract and neighborhood deprivation value. If the patient did not have a valid address at the time of the HbA1c lab result, the address at the time of their admission for labor and delivery was used. 

### Statistical Analysis

We performed descriptive bivariate analyses and mixed effects unadjusted and adjusted multivariable logistic regression models to estimate the odds of IGT per standard deviation increase in neighborhood deprivation. To account for patients living within the same census tract and therefore exposed to the same neighborhood deprivation, we used glm.cluster in the R miceadds package (v3.12-26) for all of our models to generate robust standard errors. Initial models were adjusted for age, insurance, and parity. Race and obesity were then added to the model as an additional adjustment. Given the observed attenuation and the known variation of neighborhood deprivation and IGT by race (Figure 2), we stratified these models by race. An interaction term of race and deprivation was used to determine if the neighborhood deprivation—IGT association differed by race. We performed analogous analyses for associations of deprivation with obesity.

To quantify racial disparities in IGT and obesity, we performed mixed-effects unadjusted multivariable logistic regression models and then adjusted models for age, insurance, and parity. We further adjusted for neighborhood deprivation and obesity. Given the attenuation of effect sizes with the addition of neighborhood deprivation and obesity, we performed two additional causal mediation analyses. First, we quantified the extent to which differences in neighborhood deprivation might explain Black-White disparities in IGT. Second, specifically for IGT, we quantified the extent to which differences in obesity might explain Black-White disparities in IGT. Formal mediation analysis quantifies the indirect effect of the Black vs. White race on IGT that may exist due to neighborhood deprivation or obesity (the mediator), as well as the direct effect of Black vs. White race on IGT. The direct effect includes all other potential causes of the association which may include unmeasured neighborhood-level factors such as access to green space, healthy foods, exposure to violence, intra-urban heat, noise pollution, and high police presence, as well as individual-level factors such as exposure to discrimination, food insecurity, socioeconomic position, and access to preventative care measures [28,29,30,31]. The proportion of the Black-White disparity association with IGT mediated by neighborhood deprivation or obesity is calculated with an estimate of statistical significance and degree of uncertainty (95% confidence intervals). The mediation models included individual-level adjustment of age, insurance, and parity as well as a random effect for census tract to address geographical clustering. As a sensitivity analysis, we re-ran primary analyses restricted to the individuals with measured pre-pregnancy weight (*n* = 8522). All tests were two-tailed, and *p* < 0.05 was considered statistically significant.

## 3. Results

Of the 10,642 patients in our cohort, 1132 (10.6%) had IGT; 49% self-identified as Black, 49% were Medicaid insured, and 32% were obese. Higher rates of IGT were observed among Black (15.8%) compared to White (3.1%) patients and patients of another race (9.5%) (*p* < 0.0001) as well as among obese (19.8%) compared to non-obese (6.8%) patients (*p* < 0.0001) (Table 1). Appendix A shows comparisons of patients included and excluded from missing BMI or HbA1C data; included patients were similar with respect to race and parity, but were older, more likely to be obese, and have private insurance compared to those excluded. Mean [SD] neighborhood deprivation was higher for patients with IGT (0.52 [0.13]) than those without (0.47 [0.14]) (*p* < 0.0001) (Figure 3) and was substantially higher for Black patients (0.55 [0.10]) compared to White patients (0.36 [0.11]) (*p* < 0.0001). Distributions of HbA1c and neighborhood deprivation by race are shown in Figure 2.

### 3.1. Associations of Neighborhood Deprivation with IGT and Obesity

In unadjusted models accounting for clustering of patients by census tract, increased levels of neighborhood deprivation were associated with increased odds of IGT (OR 1.44, 95%CI: 1.25, 1.67) with effect estimates per standard deviation increase in neighborhood deprivation. Similar findings were observed after adjustment for age, insurance, and parity (Table 2). The association was attenuated but remained significant with additional adjustment for race (aOR 1.15, 95%CI: 1.07, 1.24) and was further attenuated after adjustment for obesity (aOR 1.08, 95%CI: 1.00, 1.15). Adjusted models stratified by race demonstrated that neighborhood deprivation was significantly associated with IGT among Black patients (aOR 1.13, 95%CI: 1.02, 1.25). The association was not significant among White patients (aOR 1.02, 95%CI: 0.84, 1.24). However, an interaction term of race and deprivation was not significant (*p* = 0.93).

In unadjusted models accounting for clustering of patients by census tract, increased levels of neighborhood deprivation were also associated with increased odds of obesity (OR 1.78, 95%CI: 1.51, 2.11) (Table 2). This association attenuated with adjustment for age, insurance, parity, and race (aOR 1.39, 95%CI: 1.28, 1.52). In models stratified by race, the association of neighborhood deprivation with obesity was more pronounced among White patients (aOR 1.47, 95%CI: 1.25, 1.73) compared with Black patients (aOR 1.15, 95%CI: 1.07, 1.24) (interaction *p* ≤ 0.0001). Results were similar in sensitivity analyses restricted to individuals with measured pre-pregnancy weight and height (Appendix A).

### 3.2. Racial Disparities in IGT and Obesity

In models adjusted for age, insurance, and parity accounting for clustering of patients by census tract, Black patients had significantly higher odds of IGT compared with White patients (aOR 8.23, 95%CI: 6.60, 10.28) (Table 3). The association attenuated when deprivation (aOR 6.97, 95%CI: 5.53, 8.79) and obesity (aOR 5.26, 95%CI: 4.17, 6.65) were added to the model suggesting potential mediation. Formal mediation analysis adjusted for age, insurance, parity, and obesity revealed that 6.7% (95%CI: 1.6%, 11.7%) of the Black-White disparity in IGT might be explained by neighborhood deprivation. Additionally, mediation analysis adjusted for age, insurance, parity, and deprivation, revealed that 13.3% (95%CI: 10.7%, 16.7%) of the Black-White disparity in IGT may be mediated by obesity (both *p* < 0.0001).

We then modeled the Black-White disparity in obesity and found that after adjustment for age, insurance, and parity, Black patients had significantly higher odds of obesity compared with White patients (aOR 4.14, 95%CI: 3.22, 5.33). The association attenuated with additional adjustment for neighborhood deprivation (aOR 3.24, 95%CI: 2.63, 3.98) suggesting potential mediation. Formal mediation analysis suggested that 17.4% (95%CI: 12.0%, 22.4%) of the Black-White disparity in obesity may be explained by neighborhood deprivation.

## 4. Discussion

### 4.1. Principal Findings

We determined that in early pregnancy, 11% of patients have IGT, or prediabetes, and that living in a neighborhood with higher levels of deprivation was significantly associated with higher odds of this metabolic abnormality. We also identified a significant racial disparity in IGT in early pregnancy; Black patients were over five times more likely to have IGT than White patients. Through causal mediation analyses, we found that 6.7% of the Black-White disparity in IGT may be attributable to a higher deprivation in neighborhoods where Black patients live. While 6.7% may be considered a small meditation effect, it demonstrates that aspects of the environment where patients live may be associated with IGT. Further research is necessary to identify other neighborhood-level factors beyond deprivation that may contribute to IGT such as access to greenspace, walkability, and violence which have been shown to affect other aspects of pregnancy health [32,33]. As expected, obesity explained a larger proportion of the disparity (13%), which may also be partially due to differences in neighborhood conditions; deprivation explained 17% of the Black-White disparity in obesity. These findings highlight the role neighborhood conditions may play in periconception health.

### 4.2. Results

Neighborhoods can influence weight and metabolic health through multiple pathways. The neighborhood deprivation index used in this analysis has multiple area-level socioeconomic indicators and may serve as a proxy for other neighborhood characteristics such as greenspace, access to nutritious food and healthcare, violent crime, and police presence. Historic and current investment in the built environment is delineated along racial lines. As such, neighborhoods where Black people live tend to have fewer resources that improve health and an excess of exposure to factors that increase the risk of adverse health outcomes [34,35]. Meta-analyses of the built environment and diabetes in non-pregnant adults have found that higher neighborhood walkability and more greenspace are associated with lower prevalence of diabetes [36,37]. In pregnancy, lower levels of neighborhood greenness have been associated with increased odds of developing a hypertensive disorder [11]. The food environment within a neighborhood, including access, availability, and affordability of nutritious foods, impacts metabolic health. For example, in a study of non-pregnant Black patients, the incidence of diabetes was 34% higher for those living in a neighborhood with a high density of unhealthy food stores such as fast food and convenience stores [38]. Access to neighborhood healthcare resources may influence the utilization and optimization of preconception health [39]. Perceptions of neighborhood safety may increase stress, decrease physical activity, and affect the ability to consume a healthy diet, all of which can have negative metabolic consequences; together these exposures can increase allostatic load or “wear and tear” on the body leading to adverse cardiometabolic outcomes [40]. Neighborhood deprivation was significantly associated with IGT among Black patients in this cohort. In contrast, we did not detect a significant association among White patients. However, given the lack of significant interaction between race and neighborhood deprivation, we cannot conclude that neighborhood deprivation affects Black and White people differently. Nonetheless, it is likely that neighborhood deprivation influences health synergistically with individual-level exposures such as discrimination and access to preventative care, which may explain the significant association between deprivation and IGT among Black patients.

Ours is the first study to our knowledge to explore the relationship between neighborhood and early pregnancy IGT; however, our findings are consistent with other studies of neighborhood and metabolic health outside of pregnancy. Neighborhood deprivation has been associated with poorer control of diabetes outside of pregnancy [41]. The impact of changing neighborhood on metabolic health has been shown. In a study conducted through the Department of Housing and Urban Development, women living in high-poverty urban neighborhoods were randomized to receive a housing voucher and support to move to a low-poverty neighborhood. Subsequent long-term follow-up (10–15 years) found that the opportunity to move to a low-poverty neighborhood decreased the prevalence of BMI ≥ 40 kg/m^2^ by 19.1% and the prevalence of HbA1c ≥ 6.5% by 21.6% [42], demonstrating the influence of neighborhood on metabolic health.

Within our cohort, Black patients were disproportionately more likely to have IGT in early pregnancy compared to White patients which is consistent with prior reports [43]. Maternal dysglycemia contributes to adverse pregnancy outcomes in a dose-dependent relationship [44]. While it is well-established that poorly controlled pregestational diabetes and gestational diabetes are associated with pregnancy morbidities, we are beginning to understand the additional risk of early IGT in pregnancy. In an Australian population, early pregnancy HbA1c between 5.9–6.5%, excluding those treated for GDM, was associated with a 2-fold increased risk of major congenital anomalies, preeclampsia, and shoulder dystocia, and a 3-fold increased risk of perinatal death [45]. Similarly, in a Philadelphia-based cohort, early pregnancy HbA1c was associated with spontaneous preterm birth in nondiabetic patients [46]. Early pregnancy HbA1c, representing average maternal glycemia over the prior 2–3 months, is a marker of periconception metabolic health and likely influences the development of future perinatal outcomes.

### 4.3. Clinical and Research Implications

Across the United States, there are disparities in preconception health. Reproductive-age Black women have the lowest prevalence of normal weight and the highest prevalence of diabetes as compared to reproductive-age women of other races [47]. There is also a massive disparity in mortality with Black women having 2.9 times the risk of mortality as White women [48,49]. Diabetes as a cause of death in reproductive-age women has increased by 17% over the past 20 years and cardiovascular disease is the third leading cause of mortality in this group [48]. It is well established that IGT is associated with an increased risk of cardiovascular disease [50]. Our finding that neighborhood deprivation exposure differences by race may explain (mediate) a substantial proportion of the Black-White disparity in obesity suggests that in order to optimize preconception health and thus improve perinatal outcomes, clinicians and policymakers must look upstream to consider the influence of not just individual factors, but also community, neighborhood, and societal factors preventing Black people from achieving optimal pre-pregnancy health status.

### 4.4. Strengths and Limitations

Strengths of this study include the large cohort and the ability to link individual clinical data to patient addresses. A limitation of this study was that only patients with an HbA1c result were included in the analysis. While existing health system guidelines recommended universal HbA1c testing with initial prenatal labs, not all patients completed a HbA1c which may have resulted in selection bias. Another limitation is that we did not have information about the presence of hemoglobinopathies, which can impact the accuracy of HbA1c results. A large proportion of individuals in the EHR do not have weights and heights and thus were also excluded, which could have exacerbated selection bias. Patients included were similar with respect to race but were older, more likely to be obese, and less likely to be publicly insured than those excluded. The overall mediation effect was small with 6.7% of the Black-White disparity in IGT attributable to higher deprivation, however, this is only slightly less than other studies quantifying the role of neighborhood in perinatal health inequities [20,51]. Additional research into other aspects of neighborhoods that may contribute to inequities in periconceptional cardiometabolic health including walkability, green space, food access, air quality, noise pollution, and police presence is warranted.

## 5. Conclusions

In conclusion, we found that neighborhood deprivation contributes to the racial disparity in early pregnancy HbA1c, a surrogate maker of periconceptional metabolic health. Investing in neighborhoods where Black patients live may improve perinatal health equity.

## Figures and Tables

**Figure 1 ijerph-20-06175-f001:**
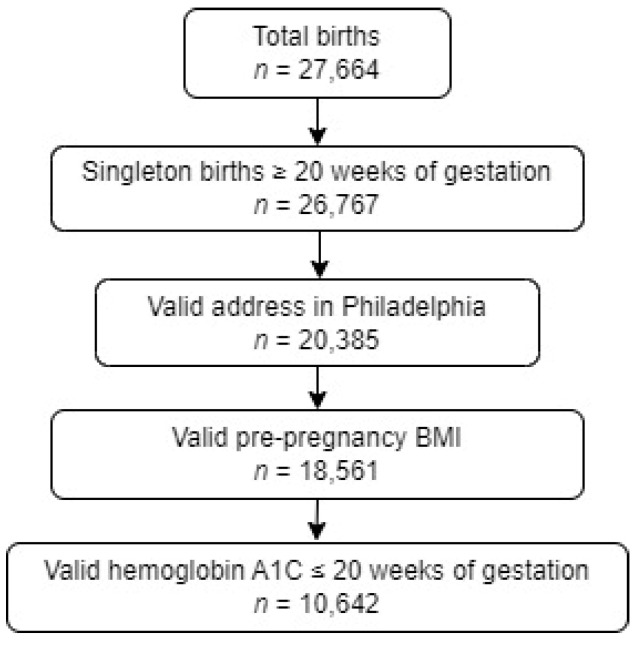
Cohort development.

**Figure 2 ijerph-20-06175-f002:**
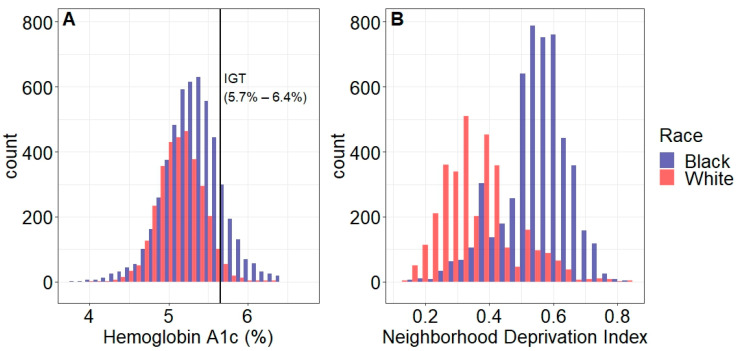
Distributions of (**A**) hemoglobin A1c and (**B**) neighborhood deprivation index among 5219 Black and 3237 White pregnant Philadelphia residents. Patients with diabetes (hemoglobin A1c ≥ 6.5%) were excluded. Vertical bar in panel A indicates hemoglobin A1c of 5.7% (cut point for impaired glucose tolerance (IGT)).

**Figure 3 ijerph-20-06175-f003:**
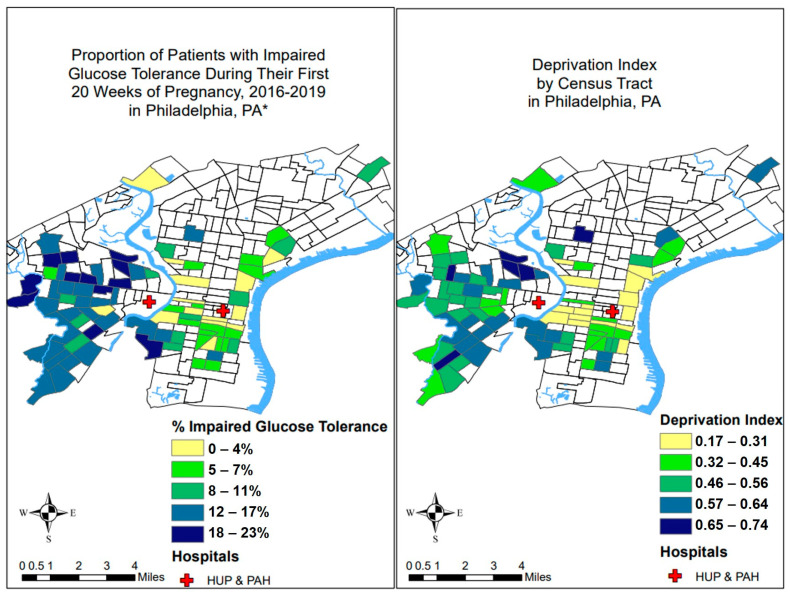
Neighborhoods with higher levels of neighborhood deprivation also had higher proportions of patients with impaired glucose tolerance in early pregnancy. HUP: Hospital of the University of Pennsylvania, PAH: Pennsylvania Hospital. * Data shown for census tracts with ≥ 50 patients.

**Table 1 ijerph-20-06175-t001:** Characteristics of 10,642 non-diabetic pregnant individuals residing in Philadelphia with glycosylated hemoglobin (HbA1c) measured during the first 20 weeks of pregnancy.

	Normal Glucose Tolerance (HbA1c < 5.7%)	Impaired Glucose Tolerance (HbA1c 5.7–6.4%)	
Characteristics	*n* (row %)	*n* (row %)	*p*-value
All	9510 (89.4)	1132 (10.6)	
Race and ethnicity			<0.001
Non-Hispanic Black	4396 (84.2)	823 (15.8)	
Non-Hispanic White	3136 (96.9)	101 (3.1)	
Another race or ethnicity	1978 (90.5)	208 (9.5)	
Insurance			<0.001
Private	5005 (91.6)	456 (8.4)	
Public	4505 (87.0)	676 (13.0)	
Pre-pregnancy BMI (kg/m^2^)			<0.001
<30 (not obese)	6795 (93.6)	463 (6.4)	
≥30 (obese)	2715 (80.2)	669 (19.8)	
Smoked during pregnancy *			0.18
Yes	554 (87.5)	79 (12.5)	
No	8827 (89.4)	1042 (10.6)	
Parity			<0.001
0	4425 (93.1)	330 (6.9)	
>0	5085 (86.4)	802 (13.6)	
	mean (SD)	mean (SD)	
Age (years)	30.0 (5.7)	31.4 (5.8)	<0.001
Pre-Pregnancy BMI (kg/m^2^)	27.2 (7.2)	32.9 (8.7)	<0.001
Neighborhood deprivation index	0.47 (0.14)	0.52 (0.13)	<0.001

* Missing smoking data *n* = 140; HbA1c: hemoglobin A1c, BMI: body mass index.

**Table 2 ijerph-20-06175-t002:** Unadjusted and adjusted associations of neighborhood deprivation with early pregnancy (<20 weeks) impaired glucose tolerance (IGT, HbA1c 5.7–6.4%), and obesity (≥30 kg/m^2^) among 10,642 pregnant, non-diabetic patients residing in Philadelphia. Models additionally stratified by race (5219 Black patients and 3237 White patients). Odds ratios presented per standard deviation increment increase in neighborhood deprivation.

Models of Neighborhood Deprivation with IGT	OR	(95% CI)
M0 = Unadjusted	1.44	(1.25, 1.67)
M1 = M0 + age, insurance, parity	1.47	(1.35, 1.61)
M2 = M1 + obesity	1.33	(1.23, 1.43)
M3 = M1 + race	1.15	(1.07, 1.24)
M4 = M1 + race + obesity	1.08	(1.00, 1.15)
**Black Patients**		
M0 = Unadjusted	1.06	(0.96, 1.17)
M1 = M0 + age, insurance, parity	1.17	(1.06, 1.29)
M2 = M1 + obesity	1.13	(1.02, 1.25)
**White Patients**		
M0 = Unadjusted	1.16	(0.97, 1.38)
M1 = M0 + age, insurance, parity	1.09	(0.91, 1.32)
M2 = M1 + obesity	1.02	(0.84, 1.24)
**Models of Neighborhood Deprivation with Obesity**		
M0 = Unadjusted	1.78	(1.51, 2.11)
M1 = M0 + age, insurance, parity	1.59	(1.42, 1.78)
M2 = M1 + race	1.39	(1.28, 1.52)
**Black Patients**		
M0 = Unadjusted	1.10	(1.03, 1.18)
M1 = M0 + age, insurance, parity	1.15	(1.07, 1.24)
**White Patients**		
M0 = Unadjusted	1.70	(1.46, 1.99)
M1 = M0 + age, insurance, parity	1.47	(1.25, 1.73)

**Table 3 ijerph-20-06175-t003:** Models of Black-White disparities in impaired glucose tolerance (IGT, HbA1c 5.7–6.4%) and obesity (≥30 kg/m^2^) among 5219 Black and 3237 White pregnant, non-diabetic patients residing in Philadelphia.

Models of IGT	OR	(95% CI)
M0 = Unadjusted	5.81	(4.80, 7.04)
M1 = M0 + age, insurance, parity	8.23	(6.60, 10.28)
M2 = M1 + deprivation index	6.97	(5.53, 8.79)
M3 = M1 + deprivation index + obesity	5.26	(4.17, 6.65)
**Models of Obesity**		
M0 = Unadjusted	4.31	(3.11, 5.98)
M1 = M0 + age, insurance, parity	4.14	(3.22, 5.33)
M2 = M1 + deprivation index	3.24	(2.63, 3.98)

## Data Availability

The data presented in this study are available on request from the corresponding author. The data are not publicly available due to ethical restrictions.

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
