# Peer review of "Neighborhood Deprivation and Racial Disparities in Early Pregnancy Impaired Glucose Tolerance"

_ijerph, 2023, doi:10.3390/ijerph20126175_

Round 1

Reviewer 1 Report

This retrospective cohort study attempts to determine a causal relationship between IGT in pregnancy and deprived neighbourhood. The data are detrived from health records and the limitations in obstaining data has been decleared by the authors. 

The authors have described a statistical method to determine the relationship between these variables and have put forth a valid argument. The method used is acceptable and the statistical model is also valid. Clearly the disproportinate variation between Blacks and Whites has some relationship with socio-economic factors and education. The scope of the study is well defined and the objectives of the study have been demonstrated convinsingly.

The Introduction gives a background to the diagnosis of pregnancy using HBA1c. It would be good to include the WHO definition of gestational diabetes ussing oral glucose torance tests to put the record straight, althouhg in this context, the authors are focussing on early pregnancy ( < 20 weeks gestation). 

Methods : This section is well written. The development of the model for the study is also well done. 

Results: Two important variables are essential to the study i.e. IGT and obesity. Obesity is missing in the title of the study. Obesity has a strong bearing to insulin resistance and consequent IGT. Additionally, the number of subjects with both IGT and obesity is not stated in the findings. 

The title for all figures should appear below the graphs and not on top.

Discussion:

This section is fairly well written. The concerns of the disparity between Blacks and Whites socio-economically is better supported by some economic indices in the geographical region. Although the authors attempt to relate the incidence of IGT and obesity to the social environment, it would be good to add the impact genetics, exercise and eating habits has in the development of type 2 diabetes mellitus.

Under implications of the study, the authors could suggest longitudinal follow up of subjects with IGT and obesity (i) to determinine if they develop T2DM and (ii) develop intervention studies to prevent development of T2DM.

Conclusions are well written.

Ethical approval has been obtained and all references are correctly written. 

Reviewer 2 Report

Interesting case study in which neighbourhood deprivation was introduced as a additional, and potentially clinically relevant, parameter to explain the prevalence of pre-diabetes in a given community/cohort during early pregnancy.

Comments:

- IGT is IGT, meaning impaired glucose tolerance (>200 mg/dl 2 h after ingesting glucose 75 g).  The American Diabetes Association labels HbA1C 5.7-6.4% as prediabetes (Standards of care in diabetes 2023, table 2.5.).  Calling this IGT is inappropriate.

- The measurement of HbA1C is subject to a large number of limitations, depending on the Hb turnover.  The turnover is altered by pregnancy itself, nutritional deficiencies (eg Fe-deprivation has been shown to increase HbA1C level in late pregnancy in diabetic women, Japanese data) and Hb variants (thalassemia, sickle cell trait).  None of this is mentioned in the manuscript, while it may be of particular importance in non-Caucasian populations.  Is there a program to screen for Hb variants in early pregnancy (this reviewer being not a US citizen)?  Are women recommended to take nutritional supplements?

- Neighbourhood deprivation is expected to show a high level of collinearity with other factors such as insurance availability, race, and perhaps even parity.  Initial models were adjusted for age, insurance and parity (line 131).  Why?  Why not analysing the contribution of each factor on its own?

- The calculation of BMI was subject to a number of inferences in a subgroup: taking 64 inches as their height, and calculating pre-pregnancy weight from weight at delivery.  Is this practice condoned by previous methodological studies?  How large was the number of persons with "inferred obesity"?

- While it is certainly worthwile to look at the neighbourhood as a driver of health, the actual living quarters are usually neglected: number of people living in the housing unit, cooking unit yes/no available, bedroom/person index, presence of pets within the housing unit, etc.  Do the authors agree?

- The same is true for self-perceived health and ability.  If you are suffering from pains, or have a mental health issue, green spaces in your neighbourhood may not appear to be a bonus in your life.  Have the authors considered adding some level of personal health/fitness/ability (eg, employed/unemployed, etc. )?

- Interactions between self-identified 'black' and 'white', but not between 'Hispanic/other/multiple/unknown'.  Why not?  Why continuing to see things in 'black and white'?? 
